# Rapid Crystallization and Fluorescence of Poly(ethylene terephthalate) Using Graphene Quantum Dots as Nucleating Agents

**DOI:** 10.3390/polym15173506

**Published:** 2023-08-22

**Authors:** Liwei Zhao, Yue Yin, Wanbao Xiao, Hongfeng Li, Hao Feng, Dezhi Wang, Chunyan Qu

**Affiliations:** 1Institute of Petrochemistry, Heilongjiang Academy of Sciences, Harbin 150040, China; xiaowanbao@126.com (W.X.); lihongfengcn@126.com (H.L.); fhbighouse2008@163.com (H.F.); jim603@163.com (D.W.); quchunyan168@163.com (C.Q.); 2Harbin FRP Institute, Harbin 150036, China; yinyue@stu.hit.edu.cn

**Keywords:** poly(ethylene terephthalate), crystallization, nucleating agent, graphene quantum dots, fluorescence

## Abstract

In this study, graphene quantum dots (GQDs) with a diameter of ~3 nm were successfully synthesized and incorporated into a poly(ethylene terephthalate) (PET) matrix to fabricate PET/GQDs nanocomposites. The impact of GQDs on the crystallization and thermal stability of the PET/GQDs nanocomposites was investigated. It was observed that the addition of only 0.5 wt% GQDs into the nanocomposites resulted in a significant increase in the crystallization temperature (peak temperature) of PET, from 194.3 °C to 206.0 °C during the cooling scan process. This suggested that an optimal concentration of GQDs could function as a nucleating agent and effectively enhance the crystallization temperature of PET. The isothermal crystallization method was employed to analyze the crystallization kinetics of the PET/GQDs nanocomposites, and the data showed that 0.5 wt% GQDs significantly accelerated the crystallization rate. Furthermore, the incorporation of GQDs into the PET matrix imparted photoluminescent properties to the resulting PET/GQDs nanocomposites. The PET crystals with GQDs as crystal nuclei and the crazes caused by defects played a vital role in isolating and suppressing the concentration quenching of GQDs. This effect facilitated the detection of defects in PET.

## 1. Introduction

PET is a highly versatile thermoplastic engineering plastic with exceptional mechanical properties, chemical resistance, heat resistance, and dimensional stability [1,2,3,4]. PET is commonly used to produce fibers, films, tapes, plastic bottles, and engineering plastics [5,6]. However, PET’s rigid chain structure and low nucleation performance results in a slow crystallization rate [7,8]. Therefore, high mold temperature and long processing time are required to produce PET-based engineering plastic components. This, in turn, results in high energy consumption, low productivity, and increased production costs, which limit the application of PET as engineering plastics.

Researchers have focused on adding small amounts (≤5.0 wt%) of heterogeneous nucleating agents [9], such as inorganic nucleating agents like silica [10,11], boron nitride [1], graphene [12,13], and clay [14,15], which significantly improve PET’s crystallization rate. However, these nucleating agents are not compatible with the PET matrix, making it difficult to disperse uniformly in the matrix. To overcome these issues, organic–inorganic hybrid nanoparticles, such as quantum dots [16], polyhedral oligomeric silsesquioxane (POSS) [17,18], and layered double hydroxide (LDHs) [19,20] and metal–organic frameworks (MOFs) [21,22] are being researched as new types of nucleating agents due to their controllable structure and excellent compatibility.

GQDs are a kind of new class of organic–inorganic hybrid nanoparticles with atomic-scale graphite planes [23,24,25], usually 0.5–2 nm thick and less than 15 nm in diameter [26]. GQDs have many hydroxyl and carboxyl groups at their edges, which allow them to be more dispersed while retaining the physical and chemical properties of graphene, making them more soluble in solvents [27]. In recent research, several studies have explored the use of quantum dots as nucleating agents to improve the crystallization properties of thermoplastics. Wang et al. [28] demonstrated a novel approach by using silica-coated CdSe QDs as nucleating agents for polyethylene terephthalate (PET) nanocomposites. The authors observed improved crystallization properties and enhanced mechanical properties in the resulting nanocomposites. Wang et al. [29] involved the preparation of graphene quantum dots (GQDs) from brown–black humic acid and the subsequent creation of poly(phenylene sulfide) (PPS)/GQDs nanocomposites via a melt blending method. The authors found that the incorporation of GQDs significantly improved the crystallinity of PPS. Gong et al. [30] introduced CdSe-ZnS QDs with varying concentrations (0–2 wt%) into poly(lactic acid) (PLA) using a simple solution casting method. The authors observed that the QDs acted as phase nucleating agents, enhancing the melt crystallization of PLA and improving its mechanical properties. Although these studies suggest that QDs can effectively act as nucleating agents to improve the crystallization properties of thermoplastics, the fluorescence properties of the resulting nanocomposites have not been fully explored. Further research in this area could yield promising results for the development of functional materials with enhanced properties.

In this study, the researchers introduced GQDs as nucleating agents into the PET matrix to prepare PET/GQDs nanocomposites with different GQDs contents. The effects of GQDs on PET’s crystallization rate and crystallization behavior, PET matrix structure, and thermal properties were analyzed. Furthermore, the fluorescence properties of PET/GQDs nanocomposites were studied and discussed. This study aims to understand how GQDs affect the crystallization behavior and thermal properties of PET and explore the potential of GQDs as nucleating agents for PET-based engineering plastics.

## 2. Experiments

### 2.1. Materials

Pure PET was provided by the School of Chemical Engineering and Chemistry, Harbin Institute of Technology (Harbin, China). Citric acid (CA) was purchased from Shanghai Aladdin Biochemical Technology Co., Ltd. (Shanghai, China). 1,1,1,3,3,3-Hexafluoroisopropanol (HFiP) was purchased form Langhua Chemical Co., Ltd. (Jinan, China). All of the chemicals were used without further purification.

### 2.2. Preparation of Graphene Quantum Dots (GQDs)

According to Dong’s bottom–up method [26], graphene quantum dots (GQDs) were synthesized via the pyrolysis of citric acid (CA). Initially, 2 g of CA were placed in a 50 mL beaker and heated in an oven at 200 °C. After 5 min, the CA melted, and the formation of bubbles was observed. With increasing pyrolysis time, the color of the CA melt changed from colorless to light yellow. After approximately 20 min, the CA melt turned orange, indicating the successful formation of GQDs. Upon cooling the crude GQDs product to room temperature, the pH of the solution was adjusted to 7 by adding a 1 mg/mL NaOH solution dropwise under vigorous stirring, resulting in the formation of an aqueous GQDs solution.

### 2.3. Preparation of PET/GQDs Nanocomposites and PET/GQDs Nanocomposite Films

PET/GQDs nanocomposites were prepared using a solvent mixing method. Prior to the experiment, PET pellets were dried in a constant temperature blast drying oven at 80 °C for 4 h to remove any moisture. Next, 5 g of PET pellets were dissolved in 40 mL of HFiP, followed by the addition of varying mass fractions of GQDs (0.5 wt%, 1 wt%, 3 wt%, and 5 wt% based on the weight of PET). The mixture was stirred at 50 °C for 4 h and sonicated for 6 h, and then poured into a Teflon tray for the natural evaporation of the solvent at ambient temperature. The resulting solid was dried in an oven at 80 °C for 24 h and then transferred to a vacuum oven at 80 °C for another 24 h to remove any remaining solvent. Finally, the solid was pulverized into a powder using a high-speed pulverizer. The resulting samples were labeled as PET-GQDs-0.5, PET-GQDs-1, PET-GQDs-3, and PET-GQDs-5 based on the varying amounts of GQDs.

The PET/GQDs nanocomposite films were prepared as follows: first, the PET/GQDs powder was dried in a constant temperature oven at 80 °C for 4 h to remove any residual moisture. The dried powder was then placed between two polyimide (PI) film molds, and the molds were heated to a temperature of 280–290 °C. The mixture was then subjected to hot pressing on a constant temperature heating plate for 3 min, during which any air bubbles were removed using a spatula. After hot pressing, the molds were immediately quenched in ice water, and the PI films were peeled off to obtain PET/GQDs nanocomposite films with a thickness of approximately 75 μm.

### 2.4. Characterization

The Fourier-transform infrared (FTIR) spectra of CA and GQDs were obtained using a Nicolet iS50 (Thermo Scientific, Waltham, MA, USA) infrared spectrometer. To prepare the samples, a small amount of CA was mixed with KBr and compressed into tablets for testing. Meanwhile, the GQDs were dispersed in ethanol and added dropwise to the KBr tablets. After removing the solvent in a vacuum drying oven, the samples were analyzed with a fixed number of 32 scans at a resolution of 4 cm^−1^.

GQDs were deposited onto a copper grid with a carbon support film multiple times using a micro-injector. The morphology of the GQDs was analyzed using a JEM 2100F transmission electron microscope (JEOL, Tokyo, Japan) operating at an accelerating voltage of 200 kV. Thermogravimetric analysis of the PET/GQDs nanocomposites was carried out using a TA Q500 thermogravimetric analyzer (TA, New Castle, DE, USA). Approximately 8 mg of the sample was loaded into an alumina crucible and then placed inside a platinum basket. The tensile strength of PET nanocomposite films is tested via Instron’s universal stretching machine (with a 100 N sensor). The PET film has a size of 35 mm × 2 mm, a thickness of 0.1 mm, and the tensile speed is 10 mm/min. Each sample was tested five times. The temperature was scanned from room temperature to 550 °C at a heating rate of 10 °C/min under an argon atmosphere. X-ray diffraction (XRD) patterns of PET/GQDs nanocomposites were obtained using an Empyrean Panalytical X-ray diffractometer (Malvern Panalytical, Almelo, The Netherlands). The sample film was fixed onto a smooth silicon plate to ensure a flat surface for testing, and the 2θ scanning range was set from 5° to 60°.

The crystallization behavior of PET/GQDs nanocomposites was studied using a Q20 differential scanning calorimeter (TA, New Castle, DE, USA). An amount of approximately 8 mg of the sample was placed in an aluminum crucible with a lid and then loaded into the instrument. Each sample was first heated to 280 °C at a ramp rate of 10 °C/min and held for 5 min to eliminate any thermal history. The sample was then cooled to 30 °C at a cooling rate of 10 °C/min and subsequently reheated to 280 °C at a heating rate of 10 °C/min. The degree of crystallinity (*X_c_*) of the samples was calculated using the following Equation (1):(1)$$X_{c}=\frac{{\Delta H}_{m}}{w\times {\Delta H}_{m}^{0}}\times 100\%$$
where ${\Delta H}_{m}$ represents the melting enthalpy of PET, ${\Delta H}_{m}^{0}$ represents the melting enthalpy of 100% crystalline PET, and *w* represents the weight fraction of PET in the nanocomposites. The literature has reported the melting enthalpy of 100% crystalline PET as 140 J/g [8,31].

Additionally, the non-isothermal crystallization properties of PET/GQDs nanocomposites were characterized using differential scanning calorimetry. Approximately 8 mg of the sample was loaded into a covered aluminum crucible and then placed into the instrument. The respective samples were heated to 280 °C at a ramp rate of 40 °C/min and held for 5 min to eliminate thermal history. The samples were then cooled to 30 °C with cooling rates of 5, 10, 15, and 20 °C/min, and non-isothermal crystallization was recorded. The isothermal crystallization process was carried out according to the following procedure: the specimens were first heated to 280 °C at a heating rate of 10 °C/min and kept at this temperature for 3 min to remove residual nuclei, and then quickly cooled to the selected isothermal crystallization temperatures and kept for different times [1]. The entire experiment was conducted under a nitrogen atmosphere with a gas flow rate of 50 mL/min. The fluorescence excitation and emission spectra of GQDs and PET/GQDs nanocomposites were measured using an F-7100 fluorescence spectrophotometer (HITACHI, Tokyo, Japan).

## 3. Results and Discussion

The structures of GQDs were characterized using FTIR and TEM. The prepared GQDs were first analyzed using FTIR, and the results are presented in Figure 1a. The broad absorption band observed at 3000–3500 cm^−1^ is attributed to the -OH stretching vibration [26,32], while the two absorption bands at 1708 and 1637 cm^−1^ correspond to the stretching of C=O and C=C, respectively [33]. These observations suggest that the carbonization of citric acid to form GQDs retained a large number of carboxyl groups, in addition to the C=C bonds. The abundance of carboxyl functional groups enhances the hydrophilicity and stability of GQDs in aqueous systems, while also facilitating their compatibility and dispersion in the polymer matrix. The TEM images in Figure 1b–d show that the GQDs are uniform in size and monodisperse, with diameters ranging from 1.2 to 4.2 nm and an average diameter of 2.8 nm. High-resolution TEM (Figure 1d) reveals lattice fringes in the GQDs, and the measured lattice spacing is 0.24 nm, corresponding to the (1120) crystal phase of graphite [34].

Figure 2a shows the ultraviolet excitation and emission spectra of GQDs. It can be observed from the excitation spectrum that the emission peak wavelength is 457 nm when the excitation wavelength is 365 nm. Figure 2b,c illustrates that the GQDs solution appears colorless and transparent under natural light while emitting bright blue light under a 365 nm ultraviolet light source. This signifies the excellent photoluminescence properties of GQDs. Moreover, the symmetrical emission spectrum along the peak position axis indicates that the emission wavelength of GQDs is independent of the excitation wavelength, demonstrating excitation independence [35].

The impact of GQDs on the crystallization behavior of PET was investigated using non-isothermal DSC analysis, and the corresponding DSC curves are presented in Figure 3, Appendix A. The DSC data are summarized in Table 1.

Figure 3 illustrates the cooling curves of PET nanocomposites. As the amount of GQDs increased from 0 to 0.5 wt%, the crystallization temperature (T_c_, peak temperature) of PET/GQDs nanocomposites increased significantly from 194.3 °C for pure PET to 206.0 °C for PET-GQDs-0.5. Meanwhile, the half-peak width of the crystallization peak became narrower and higher, indicating that PET required a narrower temperature range to crystallize. These findings suggest that GQDs greatly enhance the crystallization rate of PET by acting as a nucleating agent for heterogeneous nucleation during the melt-cooling process. However, when the GQDs content was further increased, the crystallization temperature began to decrease gradually, although it was still higher than that of pure PET. Moreover, the crystallization peak was still more symmetrical, indicating that while GQDs increased the crystallization rate and crystallization temperature of PET, excessive amounts of GQDs had a negative effect on PET crystallization. The correlation between the relative crystallinity and crystallization time of PET/GQDs nanocomposites is presented in Appendix A. Overall, the results suggest that a small amount of GQDs can be used as a nucleating agent to improve the crystallization properties of PET by facilitating heterogeneous nucleation during the melt-cooling process. Moreover, a lower concentration of GQDs is sufficient to promote PET heterogeneous nucleation.

Figure 4 depicts the results of the second heating scan of PET/GQDs nanocomposites with a heating rate of 10 °C/min. Table 2 summarizes the pertinent data. During the second heating process, PET and PET/GQDs nanocomposites exhibited no discernible glass transition process due to their higher crystallinity resulting from the prior cooling rate of 10 °C/min. The high crystallinity limited the change in the nanocomposite’s heat capacity during glass transition, making it difficult to observe the step transition. Moreover, due to the relative sufficiency of the melt’s crystallization during the first cooling, it was challenging to observe the cold crystallization peak during the second heating.

It is evident from Figure 4 that all PET/GQDs nanocomposites presented double melting peaks in the DSC melting endothermic peaks, whereas pure PET exhibited only one melting peak around 249.7 °C. Despite undergoing the same thermal history, only PET/GQDs nanocomposite samples had double melting peaks, which may be due to the nucleating agents in the nanocomposites leading to a small number of imperfect crystals. The initial crystal growth of PET is a rapid process due to the presence of more nucleating agents. However, when the crystal grows to a certain extent, its further growth is restricted by the neighboring crystals, making it difficult for all crystals to form relatively perfect crystals. Thus, double melting peaks were observed in the second heating curve of PET/GQDs nanocomposites.

During the second heating process, although the PET/GQDs nanocomposites showed obvious double peaks, the higher melting point (T_m2_) of the nanocomposites did not decrease significantly when the GQDs content was lower. The T_m2_ decreased only from 249.7 °C in pure PET to 248.7 °C in PET-GQDs-0.5, and the crystallinity also increased from 33.4% to 35.9%, indicating that GQDs can increase not only the crystallization temperature but also promote crystal growth. However, as the GQDs content increased further from 1 wt% to 5 wt%, the melting temperature of PET nanocomposite doublets decreased significantly, and the crystallization enthalpy also decreased to some extent, although it was still higher than that of pure PET. These findings indicate that while too much GQDs cannot further increase the crystallization temperature of PET, GQDs promote the formation of PET molecular crystallites, leading to higher crystallinity than that of pure PET. Crystallization temperature (T_c_) and melting temperature as a function of the GQDs content are presented in Appendix A. The non-isothermal crystallization activation energy of PET/GQDs nanocomposites are shown in Appendix A.

To achieve a better understanding of the nucleation mechanism of GQDs in the PET matrix, the melting process of PET/GQDs nanocomposites during the second heating scan was subjected to peak fitting analysis. This analysis describes the proportion of melting peaks with different melting points in the overall melting peak, which is presented in Figure 5. It is observed that the addition of GQDs to the PET matrix results in nanocomposites with two melting peaks: one with a lower melting point (P_low_) and the other with a higher melting point (P_high_). This is due to the introduction of nucleating agents, which leads to an increase in imperfect crystallization and the appearance of P_low_. When the addition of GQDs is 0.5 wt%, the height of P_high_ in PET-GQDs-0.5 is higher than that of pure PET, and T_m2_ is smaller than that of pure PET. Although T_m2_ has decreased, the peak value of P_high_ has increased. With the further increase in GQDs content, the melting temperature of P_low_ gradually decreases, and the proportion of P_low_ in the total melting peak of nanocomposites increases. The total crystallinity of the nanocomposites exhibits some improvement compared to pure PET, indicating that the increase in GQDs content contributes to the crystallinity of PET throughout the crystallization region. These findings suggest that a small amount of GQDs as nucleating agents can increase the crystallization rate of PET and form imperfect crystals in the PET matrix, leading to the formation of double melting peaks in the nanocomposites. However, with the further increase in GQDs content, the nucleation effect of GQDs becomes more apparent, resulting in an increase in the number of imperfect crystals and a significant decrease in the melting point of the PET matrix.

XRD analysis was carried out to investigate the effect of GQDs on the crystal structure of the PET matrix in PET/GQDs nanocomposites. The samples were annealed at 140 °C for 30 min, and the obtained XRD patterns were compared with that of pure PET. As shown in Appendix A, the characteristic diffraction peaks of PET crystals were observed at 17.5° (010), 22.7° (110), 25.8° (100), and 32.8° (101). Interestingly, all the XRD patterns of PET/GQDs nanocomposites exhibited the same diffraction peaks as that of pure PET, indicating that the introduction of GQDs did not alter the original crystal structure of PET.

Moreover, the XRD patterns of PET-GQDs-0.5 displayed sharper diffraction peaks on the (100) crystal plane, which suggests that the nucleation effect of GQDs remarkably enhanced the crystallization of the PET matrix. However, it is worth noting that with an increase in the GQDs content from 0.5 wt% to 5 wt%, the characteristic diffraction peaks of the XRD patterns of PET/GQDs nanocomposites became less distinct. This implies that the further increase in nanoscale GQDs content promoted imperfect crystallization, resulting in less clear diffraction peaks of the characteristic crystal planes.

The effect of GQDs on the tensile properties of PET is shown in Appendix A. The addition of GQDs improves the yield strength, fracture strength, and tensile modulus of PET/GQDs nanocomposites, which is attributed to the good dispersion (Appendix A) and enhancement of polymers via nanomaterials [36,37].When the content of GQDs is 0.5 wt%, the elongation at break of the PET nanocomposites is slightly increased, and then the elongation at break of the nanocomposites decreases with the increase in quantum dot content. In consideration of previous reports, the addition of certain nucleating agents can significantly compromise the thermal stability of the matrix. Therefore, the effect of GQDs on the thermal stability of PET can serve as a crucial indicator in assessing the performance of PET/GQDs nanocomposites. As depicted in Figure 6, the thermal stability of both PET and PET/GQDs nanocomposites was analyzed via thermogravimetric analysis under a nitrogen atmosphere. Table 3 presents the specific initial decomposition temperature, maximum decomposition temperature, and carbon residue. As can be observed from Table 3, the initial decomposition temperature of PET/GQDs nanocomposites (T_5%_ and T_10%_, referring to the temperature at which the weight loss reaches 5% and 10%, respectively) initially rises and then falls with an increase in GQDs content. For PET-GQDs-0.5, the T_5%_ and T_10%_ values are 410.1 °C and 421.8 °C, respectively, which are 4.9 °C and 6.8 °C higher than those of pure PET (the T_5%_ and T_10%_ of pure PET are 405.2 °C and 414.8 °C, respectively). This indicates that the addition of a small amount of GQDs into the PET matrix restricts the movement of molecular chains and enhances the thermal stability of the nanocomposites. Nevertheless, as the GQDs content is further raised, the initial decomposition temperature of PET/GQDs decreases. This can be attributed to the lower thermal stability of GQDs that contain a considerable number of organic groups, such as hydroxyl and carboxyl groups, causing the organic groups in GQDs to decompose before PET during heating.

It is worth mentioning that after the addition of GQDs, the T_max_ of the nanocomposites is somewhat elevated. This is because as the temperature increases, although both PET and GQDs are subject to degradation, the degradation of GQDs mainly involves the degradation of carboxyl and hydroxyl groups and the carbonization of quantum dots. This leads to the condensation of GQDs in the PET matrix to form large graphitic layers, which delays the further degradation of PET. Compared with PET, the carbon residue of PET/GQDs nanocomposites at 550 °C did not increase but decreased. This is because GQDs themselves contain a large number of organic components such as hydroxyl and carboxyl groups [26], which contribute to more weight loss in thermal stability tests; the large number of hydroxyl groups on the surface of GQDs also promoted the alcoholysis of PET.

The crystallization process of PET/GQDs nanocomposites can be investigated via the DSC method [38,39,40]. Based on the previous results, 0.5 wt% GQDs had a good crystallization effect on PET. Therefore, we took PET and PET-GQDs-0.5 as examples to study their crystallization process via the isothermal crystallization method, as shown in Figure 7. The isothermal crystallization curves of neat PET and PET-GQDs-0.5 nanocomposites are shown in Figure 7a and Figure 7b, respectively. Isothermal crystallization temperatures were selected according to the non-isothermal crystallization results [1]. For both PET and PET-GQDs-0.5, the time corresponding to the crystallization peaks increased with the increase in temperature. The crystallization peak time of PET increased from 1.61 min to 2.63 min as the crystallization temperature increased from 197 °C to 201 °C, while the crystallization peak time of PET-GQDs-0.5 increased from 0.31 min to 0.54 min as the crystallization temperature increased from 206 °C to 210 °C. The slow crystallization at high temperatures is due to the better motility of PET molecular chains at this temperature. PET-GQDs-0.5 is a heterogeneous nucleation system, and nucleation is less affected by temperature than the PET homogeneous nucleation system. Therefore, the isothermal crystallization peak of the PET-GQDs-0.5 system is narrow.

To further compare the differences in isothermal crystallization processes of the different systems, the relative crystallinity (*X_t_*) was calculated according to the reported method [1,41], as shown in Figure 7c,d. Even at higher crystallization temperatures, PET-GQDs-0.5 can achieve higher relative crystallinity in less time than PET. This shows that GQDs can effectively improve the crystallization capacity and crystallization rate of PET. Isothermal crystallization kinetic parameters were calculated to more clearly illustrate the crystallization-promoting effect of GQDs, which is calculated via the Avrami equation as follows [42].
(2)$$X_{t}=1-exp(-Kt^{n})$$
where *K* is the crystallization rate constant and *n* is the Avrami index. The fitted bilogarithmic curves are shown in Figure 7e,f, and the Avrami kinetic parameters are presented in Table 4. Each curve contains distinct linear and nonlinear parts. In the initial stages of crystallization, crystals grow naturally around the nucleus and are not affected by other crystals, thus compounding the Arrhenius formula. In the later stage of crystallization, the quality of spherulite continues to improve and the volume increases, which causes adjacent spherulite to collide, resulting in nonlinear relationships [43]. In fact, Table 4 presents isothermal kinetic parameters in the initial phase of crystallization. For both PET and PET-GQDs-0.5, the crystallization rate constant *K* decreases as the isothermal crystallization temperature increases, while PET-GQDs-0.5 exhibits a higher *K* value.

Overall, the introduction of 0.5 wt% GQDs into the PET matrix effectively promoted the crystallization of PET as a nucleating agent, increased the crystallization rate, and raised the crystallization temperature. This finding could be attributed to the extremely small particle size and good compatibility of GQDs, which enhanced the PET nucleation effect.

The solid-state ultraviolet excitation and emission spectra of PET-GQDs-5 film are shown in Figure 8. As evident from the spectra, at an excitation wavelength of 355 nm, the emission peak occurs at 450 nm, exhibiting a bright blue light. Remarkably, the introduction of GQDs into the PET matrix and subsequent fabrication of the nanocomposite films does not compromise the excellent photoluminescent properties of GQDs. Moreover, the symmetrical emission spectra suggests that the emission wavelength of PET/GQDs nanocomposites is independent of the excitation wavelength, demonstrating excitation wavelength independence.

Figure 9 presents a comparison of the fluorescence effects between PET and PET-GQDs-5 films under different conditions. As depicted in Figure 9a, PET has higher transparency under visible light compared to the amorphous PET-GQDs-5 film, which exhibits a pale yellow. This can be attributed to the inherent yellow color of GQDs, resulting in the pale yellow of the amorphous PET-GQDs-5 film. In contrast, Figure 9b shows the fluorescence effect of PET and PET-GQDs-5 films under 365 nm ultraviolet light. It is evident that pure PET lacks fluorescence, whereas PET-GQDs-5 exhibits a bright blue fluorescence, thereby demonstrating the photoluminescent characteristics of the PET-GQDs-5 nanocomposites.

To investigate the effect of crystallization on the fluorescence properties of the PET/GQDs nanocomposite films, the PET-GQDs-5 film was cold crystallized at 140 °C for 30 min, and its fluorescence was observed, as shown in Figure 9c,d. After 30 min of cold crystallization, the PET-GQDs-5 film exhibited a pale yellow with reduced transparency, as observed in Figure 9c. However, the crystallized PET-GQDs-5 film shown in Figure 9e displayed a bright blue fluorescence, with significantly higher fluorescence brightness than the amorphous PET-GQDs-5 film in Figure 9b. This indicates that crystallization enhances the fluorescence intensity of the PET/GQDs nanocomposites.

The self-quenching phenomenon of GQDs particles, caused via the interaction of high concentration when the film is amorphous, may be responsible for the fluorescence quenching between GQDs. After cold crystallization of the nanocomposite films, GQDs, serving as nucleation centers, are encapsulated by PET crystals, thereby limiting the fluorescence quenching between GQDs. The suppression of the quenching phenomenon leads to the enhancement of the fluorescence intensity of PET/GQDs nanocomposites.

Figure 9g shows a fluorescent image of the amorphous PET-GQDs-5 nanocomposite film under ultraviolet light after being bent. When compared to the image taken before bending (Figure 9f), there is no discernible change in fluorescence intensity in the non-bending region of the nanocomposite film. However, there is a significant increase in fluorescence intensity in areas that have been subjected to bending-induced defects and damage. This phenomenon may be attributed to two factors resulting from concentrated stress: the production of crazes due to stress strain, and the crystallization induced via strain orientation. Firstly, crazes usually manifest as microcracks that are approximately 10 nm in size, and GQDs within the craze–shear band become completely separated, reducing the likelihood of GQDs self-quenching. Secondly, during PET yield orientation crystallization, GQDs become encapsulated by PET crystals, thereby also inhibiting self-quenching. As a result, the defected regions of PET/GQDs nanocomposites exhibit higher fluorescence brightness.

## 4. Conclusions

In conclusion, PET/GQDs nanocomposites were fabricated using solvent blending. The incorporation of 0.5 wt% GQDs into the PET matrix enhanced the thermal stability of the nanocomposites. However, as the GQDs content increased, the thermal stability decreased due to the degradation of organic groups such as carboxyl and hydroxyl groups on the GQDs surface. The most significant finding was that the addition of GQDs greatly increased the crystallization temperature of PET, from 194.3 °C for pure PET to 206.0 °C for PET-GDQs-0.5, demonstrating the effective nucleation role of an appropriate concentration of GQDs. The isothermal crystallization method was employed to analyze the crystallization kinetics of the PET/GQDs nanocomposites. PET shows a crystallization rate constant of 0.15 min^−1^ at 197 °C, while PET-GDQs-0.5 exhibits a high crystallization rate constant of 1.5 min^−1^ at 210 °C, and the data showed that 0.5 wt% GQDs significantly accelerated the crystallization rate. Furthermore, the addition of GQDs resulted in photoluminescent properties in the PET/GQDs nanocomposites. The PET crystals acted as nucleation sites for GQDs, and crazes caused by defects played a vital role in suppressing the concentration quenching of GQDs, facilitating the detection of defects in PET.

## Figures and Tables

**Figure 1 polymers-15-03506-f001:**
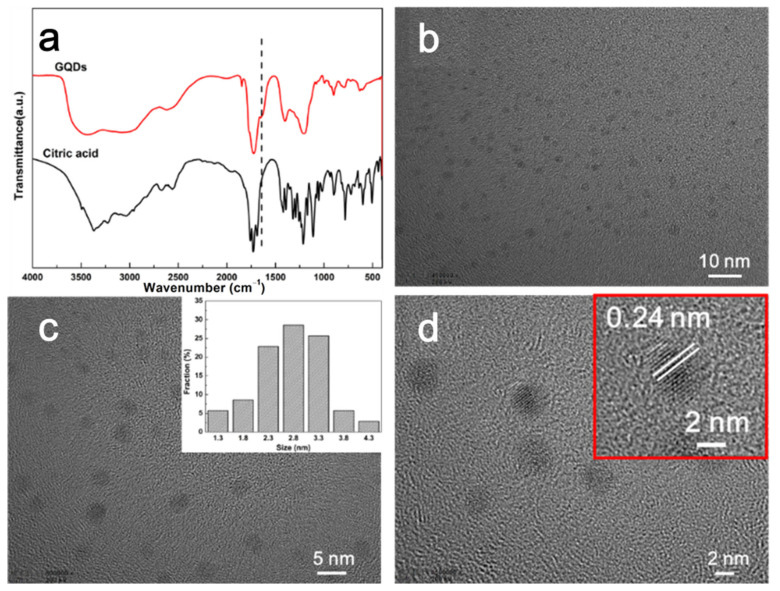
(**a**) FTIR spectra of GQDs and CA. (**b**) Low-resolution TEM image of GQDs. (**c**) TEM image of GQDs with size distribution. (**d**) High-resolution TEM image of GQDs showing the lattice spacing.

**Figure 2 polymers-15-03506-f002:**
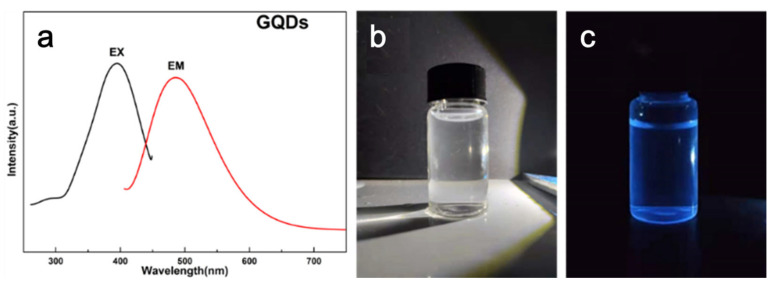
Fluorescence properties of GQDs. (**a**) Fluorescence excitation and emission spectra of GQDs, (**b**) image of GQDs solution under visible light, (**c**) image of GQDs solution under ultraviolet light.

**Figure 3 polymers-15-03506-f003:**
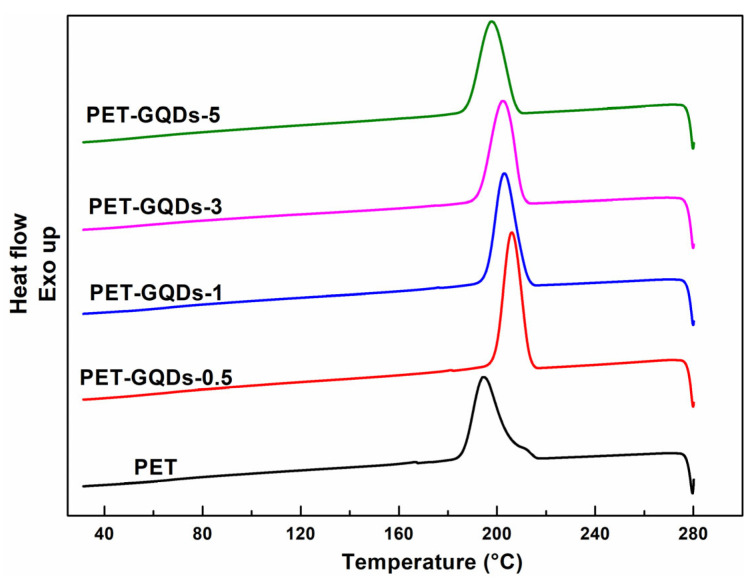
Cooling DSC scans of PET/GQDs nanocomposites after first heating.

**Figure 4 polymers-15-03506-f004:**
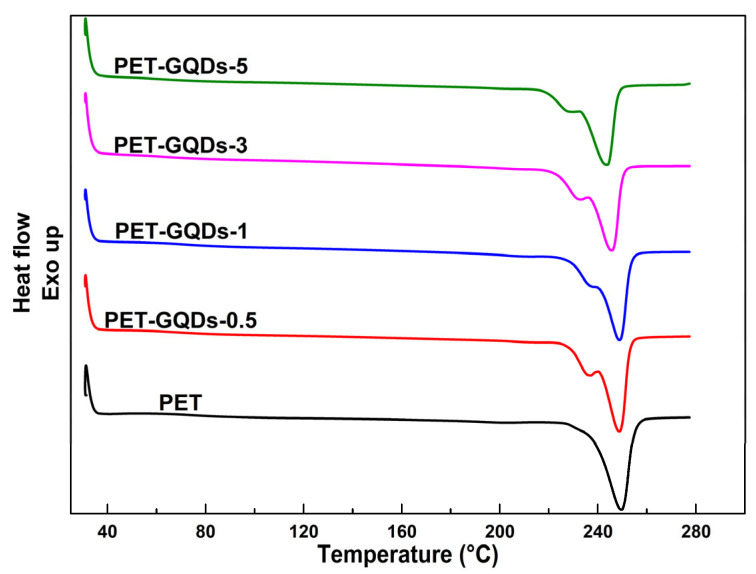
Second heating scans of PET and PET/GQDs nanocomposites.

**Figure 5 polymers-15-03506-f005:**
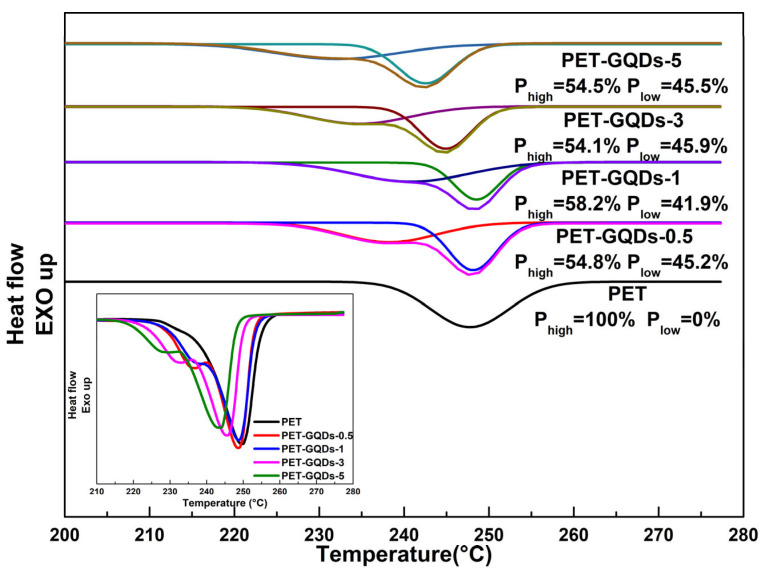
The original melting curves and the fitting curves of the PET and the PET/GQDs nanocomposites in the second heating scan.

**Figure 6 polymers-15-03506-f006:**
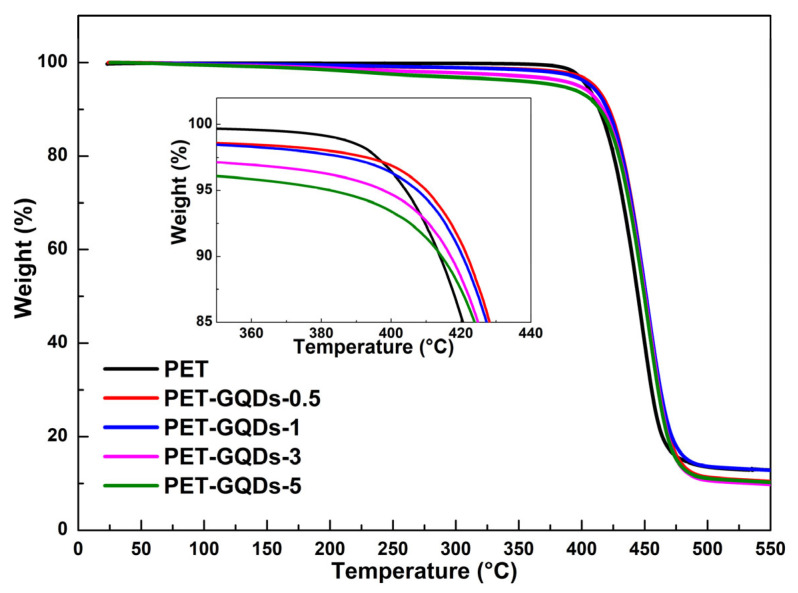
TGA curves of PET and PET/GQDs nanocomposites.

**Figure 7 polymers-15-03506-f007:**
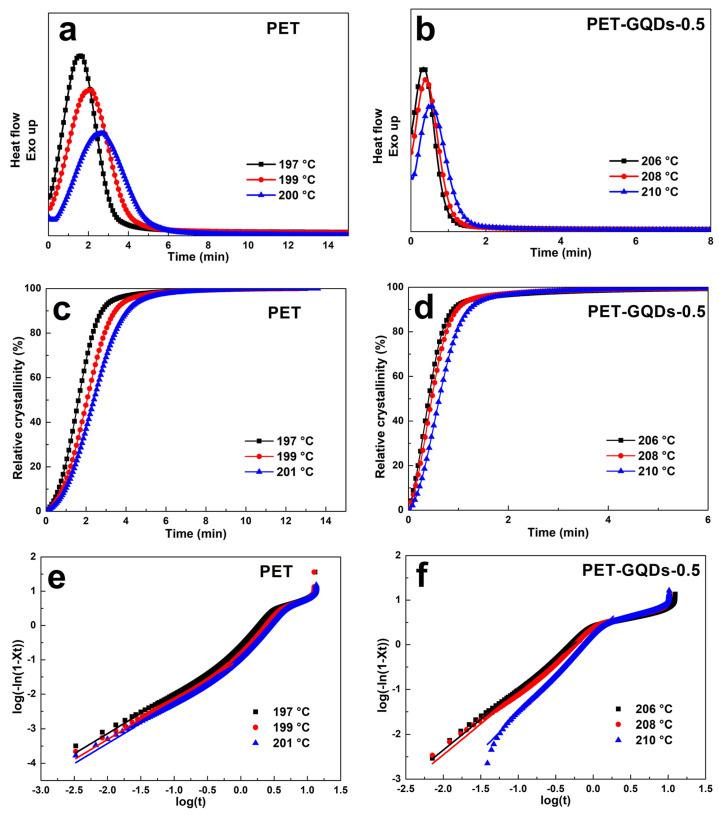
Isothermal crystallization of PET and PET-GQDs-0.5 nanocomposites. (**a**,**b**) Isothermal crystallization DSC curves, (**c**,**d**) relative crystallinity at different temperatures, (**e**,**f**) plots of log[−ln(1 − X_t_)] versus logt.

**Figure 8 polymers-15-03506-f008:**
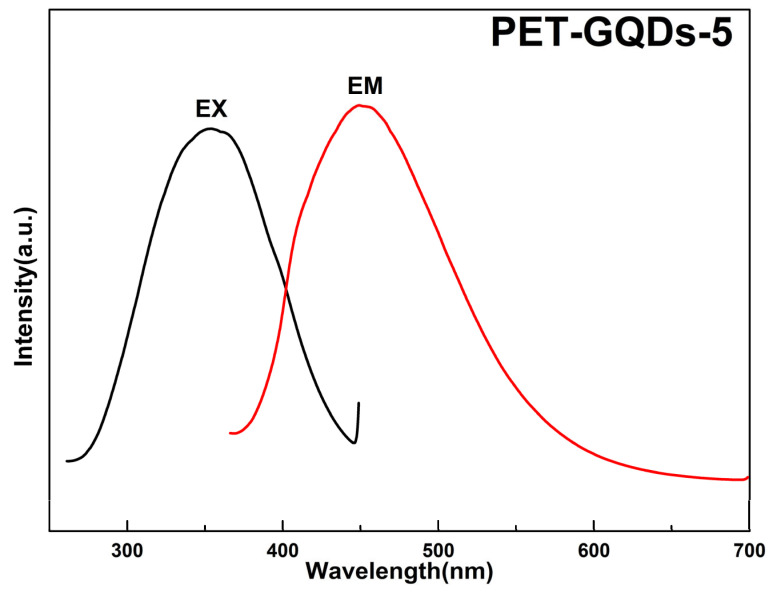
Fluorescence excitation and emission spectra of PET-GQDs-5.

**Figure 9 polymers-15-03506-f009:**
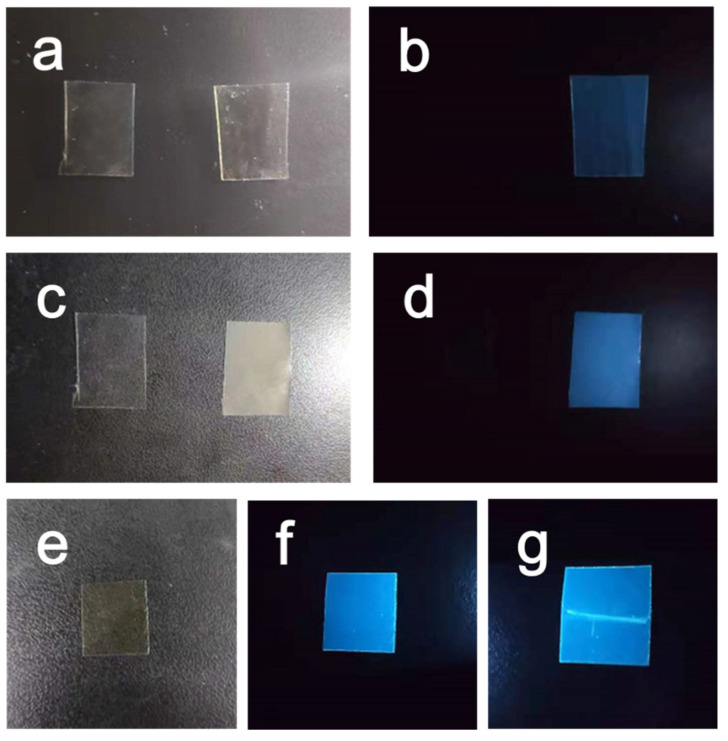
Evolution of different phase fractions for PET/GQDs nanocomposites. (**a**) Image of PET film and amorphous PET-GQDs-5 film under visible light; (**b**) image of PET film and amorphous PET-GQDs-5 film under ultraviolet light; (**c**) image of PET film and crystalline PET-GQDs-5 film under visible light; (**d**) image of PET film and crystalline PET-GQDs-5 film under ultraviolet light; (**e**) image of amorphous PET-GQDs-5 film under visible light; (**f**) image of PET-GQDs- 5 film under ultraviolet light; and (**g**) image of the PET-GQDs-5 film under ultraviolet light after stress concentration.

**Table 1 polymers-15-03506-t001:** Thermal behaviors of PET and PET/GQDs nanocomposites obtained from DSC.

Sample	Cooling after 1st Heating	
	T_c_ (°C)	ΔH_c_ (J/g)
PET	194.3 ± 0.2	52.1 ± 0.4
PET-GQDs-0.5	206.0 ± 0.1	52.3 ± 0.3
PET-GQDs-1	202.9 ± 0.1	51.9 ± 0.5
PET-GQDs-3	202.2 ± 0.2	51.4 ± 0.1
PET-GQDs-5	197.7 ± 0.3	50.4 ± 0.3

**Table 2 polymers-15-03506-t002:** Thermal behaviors of PET and PET/GQDs nanocomposites obtained from second heating scan.

Sample	T_m1_ (°C)	T_m2_ (°C)	Δ*H_m_* (J/g)	*X_c_* (%)
PET	-	249.7 ± 0.2	46.8 ± 0.4	33.4 ± 0.1
PET-GQDs-0.5	236.9 ± 0.2	248.7 ± 0.1	50.0 ± 0.6	35.9 ± 0.1
PET-GQDs-1	238.2 ± 0.2	248.8 ± 0.4	47.1 ± 0.5	34.0 ± 0.2
PET-GQDs-3	233.0 ± 0.3	245.7 ± 0.4	48.4 ± 0.2	35.6 ± 0.2
PET-GQDs-5	230.0 ± 0.4	243.7 ± 0.2	47.5 ± 0.5	35.7 ± 0.3

**Table 3 polymers-15-03506-t003:** Thermal properties of PET and PET/GQDs nanocomposites.

Sample	T_5%_ (°C)	T_10%_ (°C)	T_max_ (°C)	Residue (%)
PET	405.2 ± 0.3	414.6 ± 0.4	446.2 ± 0.4	13.2 ± 0.1
PET-GQDs-0.5	410.1 ± 0.4	421.8 ± 0.2	453.4 ± 0.5	10.4 ± 0.2
PET-GQDs-1	407.6 ± 0.2	420.5 ± 0.5	452.5 ± 0.4	12.8 ± 0.2
PET-GQDs-3	397.8 ± 0.5	417.1 ± 0.2	454.0 ± 0.3	9.9 ± 0.4
PET-GQDs-5	382.6 ± 0.2	414.5 ± 0.4	453.3 ± 0.2	10.1 ± 0.3

**Table 4 polymers-15-03506-t004:** Detailed isothermal crystallization kinetic of PET and PET/GQDs nanocomposites.

Samples	T (°C)	*n*	K (min^–1^)
PET	197	1.16 ± 0.01	0.15 ± 0.02
	199	1.15 ± 0.02	0.10 ± 0.01
	201	1.16 ± 0.01	0.08 ± 0.01
PET-GQDs-0.5	206	1.35 ± 0.01	2.34 ± 0.03
	208	1.37 ± 0.02	1.93 ± 0.01
	210	1.70 ± 0.03	1.50 ± 0.01

## Data Availability

Data is available from the author on request.

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
