# Peer review of "Rapid Crystallization and Fluorescence of Poly(ethylene terephthalate) Using Graphene Quantum Dots as Nucleating Agents"

_polymers, 2023, doi:10.3390/polym15173506_

Round 1

Reviewer 1 Report

The authors indicate that they have used the "Jeziorny method" to analyze the crystallization kinetics. Although the Jeziorny method is very popular, it should be avoided as invalid and useless. See the detailed analysis published in this same journal [1]. Therefore, I consider that a paper using Jeziorny's method to analyze the cold crystallization of a polymer cannot be accepted for publication. Kinetic analysis of cold crystallization is complex, but there are several methods to cope with this situation, for example see references [2,3].

The authors have used a peak fit to decompose the crystallization into two processes. What function has been used to fit the kinetics? What is the confidence of this fit?

Finally, how does the addition of GQDs affect the crystal size and, more importantly, the mechanical properties?

Minor comments:

In the abstract, the authors have omitted to indicate that the crystallization kinetic analysis applies to cooling experiments. Because of this omission, the message that the addition of GQDs results in a significant increase in crystallization temperature can be interpreted the other way around, that the addition of GQDs obstructs nucleation or/and growth so that the crystallization temperature increases.

What is crystallization temperature? Crystallization in polymers occurs over a range of temperatures. The temperatures used to characterize crystallization are the onset, endset and peak temperatures. The authors use the peak temperature as the crystallization temperature. This detail should be mentioned

In page 1 there are two typographical errors: “agents6” and “boron nitride1”.

In page 3. The acronym  XRD should be introduced the first time that the authors mention X-ray diffraction.

In page 4: superscript missing in “3000-3500 cm-1”.

Page 9. What is WXRD?

References:

[1] Sergey Vyazovkin, Polymers 2023, 15, 197. https://doi.org/10.3390/polym15010197

[2] S. Vyazovkin, N. Sbirrazzuoli, Macromol. Rapid. Comm. 25 (2004) 733e738. https://doi.org/10.1002/marc.200300295

[3] Farjas et al, Polymer 120 (2017) 111e118. https://doi.org/10.1016/j.polymer.2017.05.053

Author Response

Please see the attachment for a detailed response.

Reviewer 2 Report

In this work the authors present the use of graphene quantum dots as nucleating agents for PET. The description of the synthesis and preparation of the materials is well explained. 0.5 wt% is the concentration which has the highest nucleating effect, furthermore they study the crystallization kinetics by the Jeziorny method. Finally they show the photoluminescent effect of QDs on PET.

However, there are some issues that should be taken into account before publication.

- More references should be added in the discussion of the results.

- In order to better observe the changes in the crystallization temperature, cold crystallization temperature and melting temperature, it is suggested to the authors to add a graph, where they present the values of these temperatures as a function of the GQDs content.

- The authors mention that the ideal concentration of GQDs in this case is 0.5 wt%, however no results are presented using a lower concentration.

- The authors studied the crystallization kinetics by Jaziorny's method, is it possible to observe the nucleating agent behavior of GQDs by isothermal crystallization?

- It is suggested to send Figure 6 to the supplementary material.

- What is the effect of having QDs in PET in terms of exiting and emitting fluorescence and what is the reason?

- Figure 10 shows how the PET samples with 5 wt% of GQDs look under UV light, however this is not the concentration that is shown to have the best fluorescence effect.

Author Response

(The authors gave the same response as above.)

Reviewer 3 Report

The manuscript discusses the synthesis of graphene quantum dots (GQD) and fabrication of nanocomposite with the Poly(ethylene terephthalate) (PET). The resultant material thermal and structural properties were investigated. Result are attempted to establish a relation between crystallization temperature increment by addition of GQD's in PET by acting as a nucleating agent. The manuscript is largely well written but has several major aspects which needs to be clarified for better understanding among the targeted readership:

1. The introduction of the manuscript is not focused on the targeted novelty of the manuscript. Please explain: (a) Why GQD's are better (or expected to be a choice of material over other options) to prepare a nanocomposite; (b) In case of PET the issue is (as per paragraph 1) is slow crystallization, but later the introduction mentions increased thermal stability etc., but it is not clear how authors envisage the use of GQD for accelerated crystallization. Kindly clarify.

2. (a) Does GQD's exhibit agglomeration in the resultant nanocomposite? (b) How did authors ensured that GQD's are well dispersed in the polymer matrix.

3. Please include the standard deviation/error-bars for the data in the Table to show the data reproducibility. 

4. Did authors observed any surface oxidation of high surface area GQD's? If yes, please share results and if no, please explain.

5. In Table 1 why did authors used data from first heating cycle despite the fact of mentioning as "eliminate thermal history in the sample" in the  section 2.4? It is suggested to share the data for the crystallization behavior from the second heating and cooling cycle.  

6. In Table 1: If cycle 1 is reliable then why ΔHm for PET-GQD's-3 is highest, while in second cycle Table 2 it is highest for  PET-GQD's-0.5. Please explain why?

7. Figure 5 clearly shows that the onset degradation temperature (or degradation starting temperature) has decreased after addition of the GQD's. This temperature may be existing between 150-200 C as per the TGA data in Figure 5. If that is the scenario, (a) how the DSC data is reliable, as it is measured well above 200 C? (b) In the Introduction section as per reference #29, the thermal stability of nanocomposite was increased. (b) Please explain why in this research the onset degradation temperature has lowered. 

English is largely acceptable in the manuscript. However, I suggest to follow spelling check to avoid minor mistakes. Such as:

1. In Abstract, the 0.5% of GQD should be clarified as wt% or vol% etc. Though in manuscript it is clear to be wt% but it is recommended to be unified.

2. In Figure 1(d) caption, the spelling for "lattice" should be corrected. 

Author Response

(The authors gave the same response as above.)

Round 2

Reviewer 1 Report

The authors have satisfactorily addressed all my observations and comments, and I therefore consider that, in its present state, it can be accepted for publication without modification.

Author Response

Thank you very much.

Reviewer 3 Report

Authors attempted to respond the raised questions and concerns. Manuscript has been significantly improved. However, several responses are not satisfactory and therefore I would like to ask authors to kindly improve the following sections:

1. With reference to Q#2 from a reviewer which was: "(a) Does GQD's exhibit agglomeration in the resultant nanocomposite? (b) How did authors ensured that GQD's are well dispersed in the polymer matrix." The response to this question is not satisfactory. It is not impossible that nano-sized particles does not significantly agglomerate to result a composite material rather than nanocomposite. Authors used SEM image with scale bar in micrometers which is 1000x larger than the target particles itself and therefore hard to be seen. The choice of right technique is crucial. I suggest authors to attempt either SEM with greater magnification or another technique to justify the same. 

2. Regarding Q#7 which was as follows: "Figure 5 clearly shows that the onset degradation temperature (or degradation starting temperature) has decreased after addition of the GQD's. This temperature may be existing between 150-200 C as per the TGA data in Figure 5. If that is the scenario, (a) how the DSC data is reliable, as it is measured well above 200 C? (b) In the Introduction section as per reference #29, the thermal stability of nanocomposite was increased. (answered already)(b) Please explain why in this research the onset degradation temperature has lowered." The response contradicts the objectives of this research. If GQD are undergoing pyrolysis at 280 C then (i) Why this temperature was selected? ; (b) If pyrolysis has occurred is it more appropriate to state that this research establishes a partially pyrolyzed/degraded GQD's composite preparation?; (c) If the answer to (b) is yes, then may be authors need to determine the PL properties evaluation before and after composite preparation. The difference in the PL intensity may be correlated with the extent of degradation. Please explain. 

English has been significantly improved. 

Author Response

(The authors gave the same response as above.)

Round 3

Reviewer 3 Report

Authors have responded to the questions satisfactorily.

I suggest authors to avoid using both terminology "composite" and "nanocomposite" in this article. It is important to categorize the material if it is a composite or nanocomposite and the unify it through-out the the manuscript.

English has been improved significantly. Spelling check is recommended.

Author Response

Dear reviewer,

Thank you very much for your comments and suggestions on our revised manuscript. I think these valuable comments are very helpful to improve our manuscript. The response to the comment is as follows.

Q: I suggest authors to avoid using both terminology "composite" and "nanocomposite" in this article. It is important to categorize the material if it is a composite or nanocomposite and the unify it through-out the manuscript.

A: Thank you very much for this comment.

We believe that the terminology “nanocomposites” is applicable to our manuscripts. Thus, in the revised manuscript and in the revised Supporting information, all "composite" were replaced by "nanocomposites". The revised part is highlighted in yellow.